# *Salmonirosea aquatica* gen. nov., sp. nov., a Novel Genus within the Family *Spirosomaceae*, Was Isolated from Brackish Water in the Republic of Korea

**DOI:** 10.3390/microorganisms12081671

**Published:** 2024-08-14

**Authors:** Kiwoon Baek, Sumin Jang, Jaeduk Goh, Ahyoung Choi

**Affiliations:** Biological Resources Research Department, Nakdonggang National Institute of Biological Resources (NNIBR), Sangju 37242, Republic of Korea; backy8575@nnibr.re.kr (K.B.); sum1ne@nnibr.re.kr (S.J.); jdgoh@nnibr.re.kr (J.G.)

**Keywords:** *Salmonirosea*, *Salmonirosea aquatica*, novel genus, novel species, genomic analysis, taxonomy, brackish water

## Abstract

A Gram-stain-negative, obligately aerobic, non-motile, rod-shaped bacterial strain designated SJW1-29^T^ was isolated from brackish water samples collected from the Seomjin River, Republic of Korea. The purpose of this study was to characterize strain SJW1-29^T^ and determine its taxonomic position as a potential new genus within the family *Spirosomaceae*. The strain grew within the range of 10–30 °C (optimum, 25 °C), pH 5.0–10.0 (optimum, 7.0), and 1–4% NaCl (optimum, 3%). Phylogenetic analysis based on the 16S rRNA gene revealed that strain SJW1-29^T^ belongs to the family *Spirosomaceae* and is closely related to *Persicitalea jodogahamensis* Shu-9-SY12-35C^T^ (91.3% similarity), *Rhabdobacter roseus* R491^T^ (90.6%), and *Arundinibacter roseus* DMA-K-7a^T^ (90.0%), while the similarities to strains within the order *Cytophagales* were lower than 90.0%. The genome is 7.1 Mbp with a G+C content of 50.7 mol%. The use of genome-relatedness indices confirmed that this strain belongs to a new genus. The major polar lipid profile consisted of phosphatidylethanolamine, and MK-7 was the predominant menaquinone. The predominant fatty acids were summed feature 3 (C_16:1_ *ω*7*c* and/or C_16:1_ *ω*6*c*), iso-C_15:0_, iso-C_17:0_ 3-OH, and C_16:0_, representing more than 80% of the total fatty acids. The phenotypic, chemotaxonomic, genetic, and phylogenetic properties suggest that strain SJW1-29^T^ represents a novel species within a new genus in the family *Spirosomaceae*, for which the name *Salmonirosea aquatica* gen. nov., sp. nov., is proposed. The type strain of *Salmonirosea aquatica* is SJW1-29^T^ (=KCTC 72493^T^ = NBRC 114061^T^ = FBCC-B16924^T^).

## 1. Introduction

The family *Spirosomaceae*, established by Larkin and Borrall [1], belongs to the order *Cytophagales* within the phylum *Bacteroidota* [2]. Based on genome-scale data, several genera previously classified under the family *Cytophagaceae* were reclassified into the family *Spirosomaceae* [3]. At present, the family *Spirosomaceae* comprises 26 genera with validly published names (LPSN, accessed on 25 April 2024) [3], with *Spirosoma* as its type genus. Members of the family *Spirosomaceae* have been isolated from diverse habitats, including freshwater sources, marine environments, soil, rhizosphere, and plants. These bacteria are Gram-negative, rod-shaped, non-spore-forming, and frequently pigmented [4]. The predominant menaquinone is MK-7, and the major polar lipid is phosphatidylethanolamine.

Members of the family *Spirosomaceae* exhibit significant potential in various environmental and biotechnological applications. They play a critical role in the biodegradation of pollutants, making them valuable for bioremediation of soil and water environments [5,6]. They contribute to soil health improvement through symbiotic relationships with plants and are involved in the production of antibiotics and enzymes, which have industrial and pharmaceutical applications [7,8]. Furthermore, *Spirosomaceae* are capable of synthesizing biopolymers, which are useful in developing biodegradable materials [9,10,11]. Additionally, *Spirosomaceae* serve as model organisms for studying microbial adaptation to environmental stressors [12].

The river microbiome, particularly that of the family *Spirosomaceae*, is influenced by various environmental factors. These factors include nutrient availability, water temperature, pH levels, salinity, and anthropogenic activities such as pollution and urbanization. In the Seomjin River, the microbial community structure is shaped by both natural and human-induced changes, affecting the ecological balance and health of the river system. Studies have shown that variations in nutrient concentrations, temperature, and pH significantly impact microbial diversity and activity in river ecosystems [13,14]. Additionally, urbanization and pollution have been observed to alter microbial communities, potentially leading to shifts in ecological functions and overall river health [15,16].

Key indicator species in river ecosystems include various bacteria, algae, and invertebrates, which help assess the ecological status and water quality [17]. In particular, the representative organism of the Seomjin River is *Liobagrus somjinensis*, which shows very high numbers compared with other regions across the country. Conservation strategies for the Seomjin River focus on protecting key species and maintaining the river’s ecological integrity through habitat restoration, pollution control, and sustainable land-use practices [18,19]. Urbanization impacts the hydrology and ecology of rivers by altering water flow, increasing sediment load, and introducing pollutants, which affect microbial communities and the overall health of the river [20,21]. Long-term trends in river ecosystem changes indicate potential consequences such as loss of biodiversity, altered nutrient cycling, and compromised ecosystem services, emphasizing the need for effective management practices [22,23].

This study presents strain SJW1-29^T^, a novel member of the family *Spirosomaceae* isolated from brackish water in the Seomjin River, Republic of Korea. Through phenotypic, genotypic, chemotaxonomic, and phylogenetic analyses, SJW1-29^T^ is proposed as a novel species of a new genus within the family *Spirosomaceae*. The aim of this research is to enhance our understanding of the ecological roles and potential applications of *Spirosomaceae* in river ecosystems, providing insights into their contributions to environmental health and biotechnological innovations.

## 2. Material and Methods

### 2.1. Isolation and Ecology

A sample was collected from brackish water in the Seomjin River, Republic of Korea (35°00′26.3″ N, 127°47′11.7″ E). The Seomjin River, a prominent river in Korea, is distinguished by its absence of estuarine banks. This unique characteristic plays a crucial role in preserving the integrity of the brackish water region, fostering recognized biodiversity, and demonstrating remarkable environmental resilience.

Isolation was carried out using a standard dilution plating method on Marine agar 2216 (MA; BD Difco, Detroit, MI, USA), followed by incubation at 20 °C for 2 weeks. Subsequently, the optimal growth temperature was determined, and working cultures of strain SJW1-29^T^ were maintained on MA at 25 °C. To preserve the strain, glycerol suspensions (20% in distilled marine broth, *w*/*v*) were prepared, and cultures were stored at −80 °C. Strain SJW1-29^T^, representing the novel strain, was deposited at the Korean Collection for Type Cultures (KCTC), the NITE Biological Resource Center (NBRC), and the Freshwater Bioresources Culture Collection (FBCC) under accession numbers KCTC 72493^T^, NBRC 114061^T^, and FBCC-B16924^T^, respectively, for further systematic research.

### 2.2. 16S rRNA Gene Phylogeny

For the 16S rRNA sequence analysis, the genomic DNA of strain SJW1-29^T^ was extracted using the DNeasy Blood and Tissue Kit (Qiagen, Albany, NY, USA) following the manufacturer’s instructions. The 16S rRNA gene sequencing was conducted using two pairs of forward and reverse primers (27F: 5′-AGAGTTTGATCCTGGCTCAG-3′ and 1492R: 5′-GGTTACCTTGTTACGACTT-3′) [24]. PCR parameters comprised an initial denaturation at 94 °C for 5 min, followed by 25 cycles of denaturation at 94 °C for 30 s, annealing at 56 °C for 1 min, and extension at 72 °C for 1 min 30 s, with a final extension step at 72 °C for 5 min. Sequencing was carried out at Macrogen (Seoul, Republic of Korea) using an ABI 3730 DNA analyzer (Applied Biosystems, Thermo Fisher Scientific, Waltham, MA, USA). The amplified gene fragments underwent Sanger sequencing using four primers: 27F, 518F, 907R, and 1492R.

The obtained 16S rRNA gene sequences were compared with sequences of bacterial species with validly published names, accessible in the EzBioCloud database (www.ezbiocloud.net/eztaxon, accessed on 25 April 2024) [25]. Multiple sequence alignments were conducted using the EzEditor2 software (www.ezbiocloud.net, accessed on 25 April 2024) [26]. Phylogenetic trees were constructed utilizing the neighbor joining (NJ) [27], maximum likelihood (ML) [28], and maximum parsimony (MP) [29] methods integrated into MEGA 7.0 software [30]. The robustness of the neighbor joining, maximum likelihood, and maximum parsimony trees was verified through bootstrap analyses based on 1000 random replicates [31].

### 2.3. Genome Sequencing, Assembly and Annotation

The complete genome of strain SJW1-29^T^ was sequenced at DNALink (Seoul, Republic of Korea) using a PacBio RS II (Pacific Biosciences, Menlo Park, CA, USA) sequencing instrument with a 20 kb library. De novo assembly of the genome sequences was conducted using PacBio SMRT Analysis 2.3.0 [32]. Following assembly, contigs were annotated using the Rapid Annotation using Subsystem Technology (RAST) server v.2.0 (https://rast.nmpdr.org, accessed on 22 April 2024) [33] and the NCBI prokaryotic genome annotation pipeline [34]. To evaluate potential contamination in the genome assemblies, the Contamination Estimator by 16S (ContEst16S) tool (https://www.ezbiocloud.net, accessed on 13 March 2024) [35] was employed. Functional categorization of genes based on Clusters of Orthologous Group (COG) was performed by querying the KEGG (Kyoto Encyclopedia of Genes and Genomes) database (https://www.genome.jp/kegg, accessed on 24 April 2024) [36]. A genome map for strain SJW1-29^T^ was also generated using the EzBioCloud Server (https://www.ezbiocloud.net, accessed on 13 March 2024). The DNA G+C content of strain SJW1-29^T^ was determined based on the genome sequence.

The genomic sequence of strain SJW1-29^T^ was uploaded to the Type Strain Genome Server (TYGS), a free bioinformatics platform for a whole-genome-based taxonomic analysis (https://tygs.dsmz.de, accessed on 24 April 2024) [37]. The phylogenomic tree was generated with the FastME 2.1.6.1 software to highlight the position of each new bacterial strain among its closest relatives from the genome BLAST distance phylogeny (GBDP) [38]. In addition, we performed phylogenomic comparisons using key parameters, including average nucleotide identity (ANI) [39], average amino acid identity (AAI) [40], digital DNA-DNA hybridization (dDDH) [41], and percentage of conserved proteins (POCP) [42] values between strain SJW1-29^T^ and other closely related type strains in the family *Spirosomaceae*. These additional analyses support the novelty of the genus.

### 2.4. Morphological and Phenotypic Characterization

For phenotypic characterization, strain SJW1-29^T^ was routinely cultured on MA at 25 °C for 5 days. Cellular morphology and cell size were examined using phase-contrast microscopy (Nikon 80i, Tokyo, Japan) and transmission electron microscopy (TEM) (CM200; Philips, Amsterdam, The Netherlands). For TEM, cells were negatively stained with 2.0% uranyl acetate on a carbon-coated copper grid. Gram staining was performed using a Gram staining kit (MBcell, Seoul, Republic of Korea). Cellular motility was observed in wet mounts using the hanging drop method. Growth on MA medium under anaerobic conditions was assessed in an anaerobic chamber filled with a gas mixture.

The temperature range and optimum for bacterial growth were determined in MA at temperatures ranging from 4 °C to 50 °C (4, 10, 15, 20, 25, 30, 37, 42, and 50 °C). To determine the pH range and optimum, strains were cultivated in artificial seawater medium (ASW), as described by Choo et al. [43], supplemented with 0.5% peptone and 0.1% yeast extract, at different pH levels ranging from 5.0 to 10.0 (at intervals of 1.0 pH units). Buffers such as MES (pH 5.0–6.0), MOPS (pH 6.5–7.0), HEPES (pH 7.5–8.0), Tris (pH 8.5–9.0), and CHES (pH 9.5–10.0) were added at a final concentration of 0.05 M to maintain the pH. The requirement for and tolerance to NaCl was determined by culturing strains in NaCl-free ASW containing 0.5% peptone and 0.1% yeast extract, with different concentrations of NaCl (0–5% NaCl at intervals of 0.5%; 5.0–15.0% NaCl at intervals of 2.5%). The turbidity of each culture was monitored daily using a spectrophotometer (NovaspecPro; Biochrom Ltd., Cambridge, UK) for up to 7 days. Degradation of Tween-20 (1.0%, *w*/*v*), Tween-40 (1.0%, *w*/*v*), and Tween-80 (1.0%, *w*/*v*) was tested on MA supplemented with each component, according to the method described by Smibert and Krieg [44].

Hydrolysis of casein (10% skimmed milk, *w*/*v*) and starch (1.0%, *w*/*v*) was determined based on the formation of clear zones around colonies after applying suitable staining solutions [45]. Following the manufacturer’s instructions, other enzymatic activities were evaluated using an API ZYM kit (BioMérieux, Lyon, France). Substrate usage was determined using the API 20NE system (BioMérieux) and API 50CH (BioMérieux).

### 2.5. Chemotaxonomic Characterization

To determine the fatty acid composition, cells of strain SJW1-29^T^ were cultivated to the late exponential phase (5 days) on MA at 25 °C. Analysis was conducted using the method described by the Sherlock Microbial Identification System version 6.1 (MIDI), utilizing the TSBA6 library [46]. Respiratory isoprenoid quinones were purified by thin-layer chromatography (TLC) following the protocol outlined in Minnikin et al. [47] and analyzed via reverse-phase HPLC.

Polar lipids of strain SJW1-29^T^ were extracted from lyophilized bacterial cells and examined using two-dimensional TLC [47]. Differential spots were detected by spraying with appropriate detection reagents. All polar lipids on the TLC plates were visualized by spraying with a phosphomolybdic acid ethanol solution (Sigma-Aldrich, St. Louis, MO, USA). Specific lipids containing functional groups were identified by spraying with zinzadze reagent (molybdenum blue spray reagent, 1.3%; Sigma-Aldrich) for phospholipids, 0.2% ninhydrin solution (Sigma-Aldrich) for aminolipids, and *α*-naphthol solution for glycolipids.

## 3. Results and Discussion

### 3.1. Isolation of Strain

Strain SJW1-29^T^ was isolated from brackish water from the Seomjin River, Republic of Korea, using a standard dilution plating method on MA. After an incubation period of 2 weeks, salmon-pink-colored colonies were observed on the MA medium. The optimum temperature for growth was determined, and cultures were subsequently maintained on MA at 25 °C.

### 3.2. 16S rRNA Gene Phylogeny

The nearly full length of the 16S rRNA gene sequence of strain SJW1-29^T^ comprises 1465 nucleotides (NCBI GenBank accession number MN596019). Comparison of the 16S rRNA gene sequence with the EZBioCloud server revealed that strain SJW1-29^T^ belongs to the family *Spirosomaceae* and was most closely related to *Persicitalea jodogahamensis* Shu-9-SY12-35C^T^ (91.3% similarity), followed by *Rhabdobacter roseus* R49^T^ (90.6%) and *Arundinibacter roseus* DMA-K-7a^T^ (90.0%). However, the similarity to other strains of *Spirosomaceae* was lower than 90.0%. These values were significantly lower than the cut-off value of 94% for allocating a strain to a novel genus [48].

The phylogenetic trees (Figure 1, Appendix A) indicate that strain SJW1-29^T^ forms a distinct lineage within the members of the family *Spirosomaceae*. This distinct phylogenetic position, coupled with the low similarity values, supports the proposal of a novel genus. Phylogenetic analyses using the neighbor joining, maximum likelihood, and maximum parsimony methods consistently placed SJW1-29^T^ in a separate clade, reinforcing its unique taxonomic status.

The use of 16S rRNA gene sequencing is a foundational approach in bacterial taxonomy and phylogeny. The significant divergence observed in SJW1-29^T^’s 16S rRNA gene sequence from other members of the family *Spirosomaceae* highlights its potential novelty. Studies by García-López et al. [4] have demonstrated the importance of integrating multiple phylogenetic methods to establish robust phylogenetic relationships among novel bacterial taxa. Additionally, the application of genome-scale data, as emphasized by Oren and Garrity [2], enhances the resolution of bacterial classification beyond what 16S rRNA analysis alone can achieve. This polyphasic approach ensures a more accurate and comprehensive understanding of the phylogenetic and taxonomic positioning of novel strains like SJW1-29^T^.

### 3.3. Genome Features

The genome features of strain SJW1-29^T^ were analyzed in this study. The DNA G+C content of strain SJW1-29^T^ was 50.7%. A summary of the genome properties and statistics of strain SJW1-29^T^ is provided in Appendix A. The draft genome sequence of SJW1-29^T^, representing the novel strain, had a size of 7,065,248 bp, consisting of four contigs with a contig N50 value of 6,512,965 bp. The genome contained a total of 6297 coding sequences, 9 rRNA genes, and 40 tRNA genes (Figure 2).

Contamination was excluded by comparing a fragment of the 16S rRNA gene with the ContEst16S results. The genome sequence met the proposed minimum standards for bacterial taxonomy [49]. A RAST analysis was performed to predict the functional gene content of the strain SJW1-29^T^ genome (Figure 3). The analysis revealed the presence of genes involved in various biological processes, including carbohydrates (351 genes), amino acids and derivatives (334 genes), protein metabolism (247 genes), and cofactors, vitamins, prosthetic groups, and pigments (188 genes). A total of 5315 protein-coding genes were identified in the genome of strain SJW1-29^T^, with 5315 genes (84.4%) assigned to COG categories. The COG category assignment of genes included cell wall/membrane/envelope biogenesis (M; 6.2%, 328 genes) and inorganic ion transport and metabolism (P; 5.6%, 299 genes), whereas 45.3% of genes were classified as having unknown functions within the COGs (Appendix A).

The genome annotations and functional characterization analysis revealed that the genome sequence of strain SJW1-29^T^ contains multiple genes encoding putative polysaccharide-degrading enzymes. These include eleven glucosidases, two chitinases, four cellulases, eight xylanases, six amylases, one alginate lyase, and six pectinesterases [50].

The phylogenomic tree constructed based on TYGS analysis revealed the relationship between strain SJW1-29^T^ and closely related type strains (Appendix A). It is most closely related to species of the genus *Persicitalea* and other genera within the family *Spirosomaceae*. The analysis clearly shows that strain SJW1-29^T^ forms a distinct clade, separate from its closest related species, with a low bootstrap value of 79%, supporting its classification as a new genus.

Considering the results obtained from complete phylogenetic analyses, to confirm that strain SJW1-29^T^ is indeed a new taxon, ANI, AAI, dDDH, and POCP values of strain SJW1-29^T^ and other related members of the family *Spirosomaceae* were calculated (Appendix A and Table 1). The genome-relatedness indices, including ANI, AAI, and dDDH, showed that this strain represents a distinct species with values below the species demarcation thresholds (ANI < 95%, AAI < 95%, and dDDH < 70%) when compared with the closest type strains within the family *Spirosomaceae* [39,51,52,53,54,55,56]. Additionally, the POCP results indicated less than 66% similarity with other genera within the family *Spirosomaceae*. POCP can serve as a powerful genomic index to define the genus boundaries of prokaryotic groups [43]. However, it should also be recognized that POCP is only a genomic index. For the genus-level separation of the family *Bacillaceae*, the POCP boundary was more than 50% [57,58]. Even in this family *Spirosomaceae*, the POCP value is more than 50% between genera, but it can be judged as a different genus based on several taxonomic grounds. Since this new taxon’s most closely related species belong to the family *Spirosomaceae*, we suggest that this new genus should also be classified within this family as a member of the order *Cytophagales* within the class *Cytophagia*.

### 3.4. Morphological and Phenotypic Characteristics

The cells of SJW1-29^T^ were observed to be Gram-stain-negative and had a rod shape, ranging in size from 0.6 to 0.8 µm × 1.3 to 1.7 µm (Appendix A). Colonies formed by the strain on the MA medium appeared regular, round, and salmon pink after 5 days of incubation at 25 °C. Strain SJW1-29^T^ exhibited a 15–30 °C temperature range with an optimum growth temperature of 25 °C. It grew within a pH range of 5.0–10.0, with optimal growth occurring at pH 7.0. The strain showed tolerance to NaCl concentrations ranging from 1.0% to 5.0% (*w*/*v*), with an optimum NaCl concentration for growth of 3.0%. Analyses of differential characteristics showed that SJW1-29^T^ and related species differed in multiple characteristics, including isolation source, salinity range, the optimum temperature for growth, enzyme activities, and carbon source utilization patterns (Table 1).

The phenotypic characteristics of strain SJW1-29^T^ align with those typically observed in the family *Spirosomaceae*, such as Gram-stain negativity and rod-shaped morphology. The ability of SJW1-29^T^ to grow in a broad range of temperatures and pH levels, as well as its tolerance to varying NaCl concentrations, suggests ecological versatility, which is a common trait in environmental isolates from brackish waters. This adaptability is crucial for survival in fluctuating estuarine environments where salinity and other abiotic factors vary significantly [11].

Comparative phenotypic analysis, as shown in Table 1, reveals distinct biochemical capabilities that further substantiate the novelty of SJW1-29^T^. For instance, the strain’s unique enzymatic profile, including its ability to hydrolyze specific substrates and its distinct fatty acid composition, differentiates it from closely related species. These phenotypic traits support the classification of SJW1-29^T^ as a new genus and highlight its potential functional roles in its native habitat. The presence of multiple genes encoding polysaccharide-degrading enzymes, as indicated by genome annotation, suggests a role in the degradation of complex carbohydrates in the ecosystem, contributing to nutrient cycling and organic matter decomposition in the Seomjin River estuary [10].

**Table 1 microorganisms-12-01671-t001:** Differential characteristics between strain SJW1-19^T^ and the most closely related species of the family *Spirosomaceae*. Genus: 1, *Salmonirosea* gen. nov. (data from this study); 2, *Persicitalea jodogahamensis* (only species in the genus) [59]; 3, *Rhabdobacter roseus* (only species in the genus) [60]; 4, *Arundinibacter roseus* (only species in the genus) [61]; 5, *Telluribacter humicola* (only species in the genus) [62]; 6, *Dyadobacter fermentans* (type species in the genus) [63]; 7, *Ravibacter arvi* (only species in the genus) [64]. +, positive; −, negative; ND, data not available.

Characteristic	1	2	3	4	5	6	7
Isolation source	Brackish water	Sea water	Soil	Freshwater	Soil	Plant	Soil
16S rRNA gene similarity (%)	ref.	91.3	90.6	90.0	89.1	88.4	84.8
Genome size (bp)	7,065,248	6,413,589	6,917,964	5,828,278	6,558,731	6,967,790	5,775,379
ANI value	ref.	75.0	71.9	71.8	71.1	69.2	67.3
AAI value	ref.	78.4	70.8	74.6	70.6	67.0	63.4
dDDH value	ref.	21.0	18.5	17.5	18.2	18.8	19.9
POCP value	ref.	62.6	63.9	66.0	63.0	52.6	46.0
Colony color	Salmon pink	Pale pink	Light pink	Light pink	Pink	Yellow	Pale yellow
Cell morphology	Rods	Straight rod to curved rods	Rods	Rods	Rods	Rods	Rods
Flexirubin reaction	−	−	ND	ND	ND	+	ND
Temperature for growth (°C)	15–30	25–30	10–37	5–37	10–42	15–37	15–40
Highest NaCl tolerated (%, *w*/*v*)	<5.0	<7.0	<5.0	<4.0	<4.0	<1.5	<1.0
Genome size (Mb)	7.07	ND	ND	5.83	ND	6.97	ND
Assimilation of (API 20NE):							
Glucose fermentation	−	+	−	W	+	+	−
Urease	−	+	−	+	ND	ND	+
Enzyme activity (API ZYM)							
Alkaline phosphatase	+	−	−	+	+	ND	+
Esterase (C4), cystine arylamidase, trypsin, *β*-galactosidase	−	+	+	+	+	ND	+
Lipase (C14)	−	−	−	−	−	ND	+
*α*-chymotrypsin	+	+	+	−	+	ND	−
Acid phosphatase	+	+	+	+	−	ND	+
*β*-glucosidase	+	+	+	−	+	ND	+
*α*-mannosidase	+	−	+	+	+	ND	+
Major quinone ^†^	MK-7	MK-7	MK-7	MK-7	MK-7	MK-7	MK-7
Major polar lipids ^‡^	PE, APL, AL, L	PE, AL, L	PE, PL, AL, L	PE, AL, GL, L	AL, L	PE, PL, AL, L	PE, APL, AL
DNA G+C content (mol%) ^§^	50.7	50.8	54.0	45.7	48.9	51.5	50.6

^†^ MK, menaquinone; ^‡^ PE, phosphatidylethanolamine; APL, unidentified aminophospholipid; AL, unidentified aminolipid; GL, unidentified glycolipid; PL, unidentified phospholipid; L, unidentified lipids; ^§^ the G+C content was calculated based on the nucleotide content of the sequenced genomes.

### 3.5. Chemotaxonomic Characterization

The major cellular fatty acids of strain SJW1-29^T^ were identified as summed feature 3 (C_16:1_ *ω*7*c* and/or C_16:1_ *ω*6*c*; 55.3%), followed by iso-C_15:0_ (11.9%), iso-C_17:0_ 3-OH (8.2%), and C_16:0_ (8.0%). A comparison of the full fatty acid profiles of strain SJW1-29^T^ can be found in Table 2. The predominant respiratory quinone in strain SJW1-29^T^ was menaquinone-7 (MK-7), a common quinone found in the family *Spirosomaceae*. The major polar lipid of strain SJW1-29^T^ was phosphatidylethanolamine (PE). Additionally, one unidentified aminophospholipid (APL), one unidentified aminolipid (AL), and seven unidentified lipids (L1–7) were also detected. No glycolipid was detected in the chromatogram when sprayed with *α*-naphthol solution and incubated at 110 °C for 15 min (Appendix A).

## 4. Conclusions

The identification and characterization of *Salmonirosea aquatica* gen. nov., sp. nov. have significant implications for microbial taxonomy and environmental microbiology. This study expands the known diversity within the family *Spirosomaceae* and underscores the importance of exploring brackish water environments to discover novel bacterial taxa. The discovery of *S. aquatica* contributes to our understanding of phylogenetic relationships within *Spirosomaceae*, revealing unique evolutionary lineages due to its low 16S rRNA gene sequence similarity (<94%) with closest relatives. Additionally, genome-based analyses, including ANI, AAI, dDDH, and POCP, further support its classification as a new genus. These findings emphasize the need for continued phylogenetic studies to uncover and classify new genera and species.

Practically, *S. aquatica*’s unique phenotypic and chemotaxonomic characteristics, including its ability to degrade various polysaccharides and tolerate diverse environmental conditions, highlight its potential applications in biotechnology and environmental management. For instance, its polysaccharide-degrading enzymes could be used in bioremediation to break down complex organic pollutants in aquatic environments. Future research should explore the functional roles of *S. aquatica* in its native habitat, its interactions within microbial communities, and its contribution to nutrient cycling and ecosystem health. Additionally, biotechnological studies can investigate the potential applications of its enzymes in industrial processes and environmental cleanup efforts.

In conclusion, the isolation and characterization of *Salmonirosea aquatica* enrich our understanding of microbial diversity and open new avenues for applied research in environmental biotechnology and ecosystem management. This study lays the groundwork for future explorations into the functional capabilities and ecological significance of novel bacterial taxa.

### 4.1. Description of Salmonirosea gen. nov.

*Salmonirosea* (Sal.mo.ni.ro’se.a. L. masc. n. *salmo*, a salmon; L. masc. adj. *roseus*, pink; N.L. fem. n. *Salmonirosea*, an organism with a salmon-pink color).

Cells are Gram-stain-negative, rod, and non-motile, and colonies on MA are salmon pink. Oxidase- and catalase-positive. Cells are obligately aerobic. Its major respiratory quinone is menaquinone-7 (MK-7). The major fatty acid components (>10.0%) included include summed feature 3 (C_16:1_ *ω*7*c* and/or C_16:1_ *ω*6*c*; 55.6%) and iso-C_15:0_, a major profile similar to those of its closest related strains. Major polar lipids are phosphatidylethanolamine. The type species of the new genus is *Salmonirosea*.

### 4.2. Description of Salmonirosea aquatica sp. nov.

*Salmonirosea aquatica* (a.qua′ti.ca. L. fem. adj. *aquatica*, living, growing, or found in or by water, aquatic).

Gram-stain-negative, oxidase- and catalase-positive, and aerobic. Cells are rod-shaped (0.6–0.8 µm in diameter and 1.3–1.7 in length, Appendix A). Growth occurs at 15–30 °C (optimum 25 °C), pH 5.0–10.0 (optimum pH 7.0), and in the presence of 1.0–5.0% (*w*/*v*) NaCl (optimum, 3.0% NaCl). Catalase and oxidase reactions are positive. Tween-20 and Tween-40 are hydrolyzed, but Tween-80, casein, and starch are not. Positive for esculin hydrolysis and PNPG (*β*-galactosidase) but negative for nitrate reduction, indole production, glucose fermentation, arginine dihydrolase, urease, and gelatinase (in API 20NE). With API ZYM, positive for alkaline phosphatase, esterase lipase (C8), leucine arylamidase, valine arylamidase, *α*-chymotrypsin, acid phosphatase, naphthol-AS-BI-phosphohydrolase, *α*-galactosidase, *α*-glucosidase, *β*-glucosidase, N-acetyl-*β*-glucosaminidase, and *α*-mannosidase, but negative for esterase (C4), lipase (C14), cystine arylamidase, trypsin, *β*-galactosidase, *β*-glucuronidase, and *α*-fucosidase. Substrates used include erythritol, D-arabinose, D-xylose, L-xylose, D-galactose, D-glucose, D-fructose, D-mannose, L-sorbose, arbutin, aesculin ferric citrate, salicin, D-melibiose, D-melezitose, gentiobiose, D-lyxose, D-tagatose, potassium 2-ketogluconate, and potassium 5-ketogluconate (API 50CH). The in silico G+C content of the whole genome of the type strain is 50.7 mol%.

The type strain SJW1-29^T^ (=KCTC 72493^T^ = NBRC 114061^T^ = FBCC-B16924^T^) was isolated from brackish water collected in the Republic of Korea.

## Figures and Tables

**Figure 1 microorganisms-12-01671-f001:**
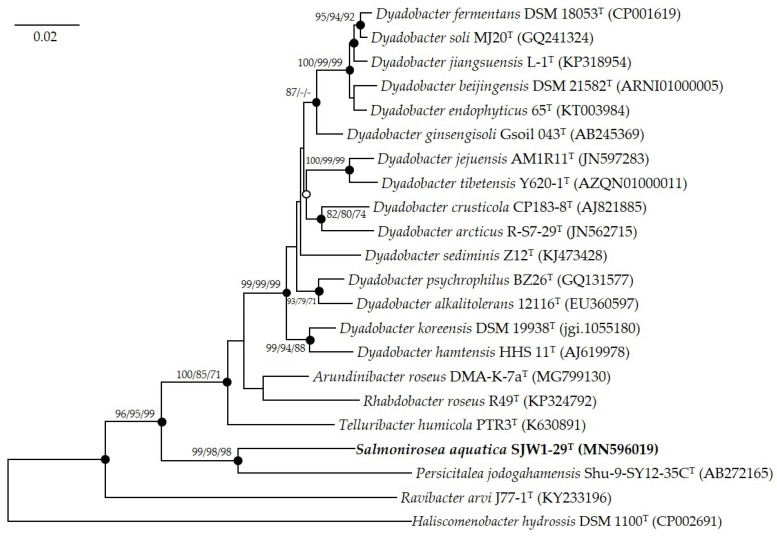
Neighbor joining tree based on nearly complete 16S rRNA gene sequences showing the phylogenetic position of strain SJW1-29^T^ with closely related taxa. Bootstrap values > 70 are indicated and based on 1000 replications (NJ/ML/MP; -, no-value-filled circles indicate that the corresponding nodes were recovered by all treeing methods. An open circle indicates that the corresponding node was recovered by the neighbor joining and maximum parsimony methods. The bold font represents the novel species identified in this study. *Haliscomenobacter hydrossis* DSM 1100^T^ (CP002691) was used as an out-group. The scale bar depicts 0.02 substitutions per nucleotide position.

**Figure 2 microorganisms-12-01671-f002:**
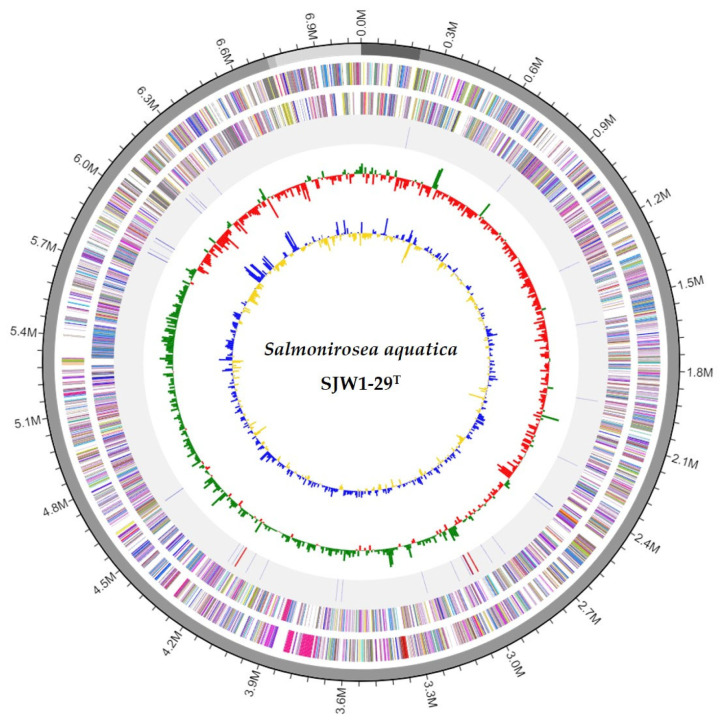
Circular map of the strain SJW1-29^T^ genome. From outside to the center: the colored bands in ring 1 represent contigs; ring 2 represents the annotated genes on the forward strand (color determined by COG category); ring 3 shows the annotated genes on the reverse strand (color determined by COG category); ring 4 displays the RNA genes (rRNAs are displayed in red and tRNAs are displayed in purple); ring 5 shows the GC skew (higher-than-average values are displayed in green, while lower-than-average values are displayed in red) and ring 6 shows the GC ratio (higher-than-average values in blue and lower-than-average values in yellow).

**Figure 3 microorganisms-12-01671-f003:**
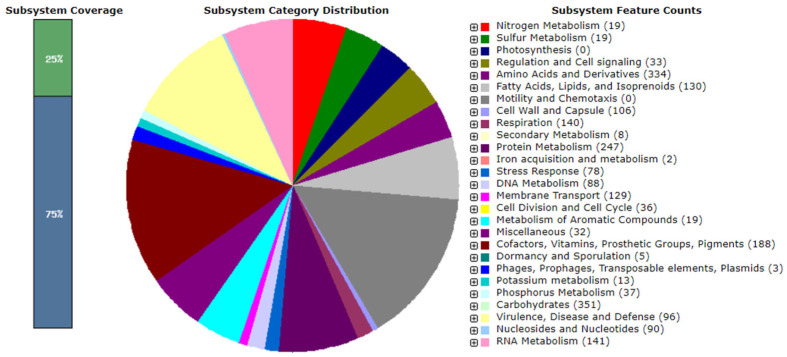
Subsystem category distribution of strain SJW1-29^T^ based on the RAST annotation server (https://rast.nmpdr.org/, accessed on 21 May 2024).

**Table 2 microorganisms-12-01671-t002:** Fatty acid profiles of strain SJW1-29^T^ and the type strains of the related genera of the family *Spirosomaceae*. Genus: 1, *Salmonirosea* gen. nov. (data from this study); 2, *Persicitalea jodogahamensis* (only species in the genus) [11]; 3, *Rhabdobacter roseus* (only species in the genus) [10]; 4, *Arundinibacter roseus* (only species in the genus) [59]; 5, *Telluribacter humicola* (only species in the genus) [60]; 6, *Dyadobacter fermentans* (type species in the genus) [61]; 7, *Ravibacter arvi* (only species in the genus) [62]. Values are percentages of the total fatty acids. tr, trace amount (<1%).

Fatty Acid	1	2	3	4	5	6	7
Saturated							
C_14:0_	1.9	-	1.0	-	tr	-	tr
C_16:0_	8.0	12.0	6.6	7.4	7.7	5.6	10.1
Unsaturated							
C_13:1_	-	2.0	1.6	-	-	1.3	-
C_14:1_ *ω*5*c*	-	-	-	-	3.7		-
C_16:1_ *ω*5*c*	1.8	6.6	17.5	14.2	13.5	15.9	6.9
Hydroxy							
C_14:0_ 2-OH	1.1	-	-	-	-	-	tr
C_16:0_ 3-OH	1.2	2.2	1.3	2.3	-	2.3	4.6
C_16:0_ N alcohol	-	-	-	-	1.1	-	-
Iso-C_15:0_ 3-OH	2.6	2.1	1.0	1.5	3.2	1.7	2.5
Iso-C_16:0_ 2-OH	-	-	-	-	-	-	1.1
Iso-C_16:0_ 3-OH	2.6	1.5	1.2	-	-	-	tr
Iso-C_17:0_ 3-OH	8.2	6.2	3.7	6.5	-	4.4	8.5
Iso-C_17:1_ *ω*9*c*	-	-	-	1.3	3.9	-	-
Branched chain							
Iso-C_10:0_	-	-	-	-	4.3	-	-
Iso-C_14:0_	1.1	-	-	-	-	-	tr
Iso-C_15:0_	11.9	31.5	21.4	16.6	13.8	19.4	9.9
Iso-C_17:0_	-	-	-	-	1.8	-	tr
Iso-C_15:0_ G	-	-	-	1.4	-		-
Anteiso-C_15:0_	1.5	1.7	-	1.7	3.5	-	tr
Summed features 3 *	55.3	29.9	40.7	41.8	39.9	44.4	52.1

* Summed features represent groups of two or three fatty acids that could not be separated using the MIDI system. Summed feature 3 comprised C_16:1_ *ω*7*c* and/or C_16:1_ *ω*6*c*.

## Data Availability

The GenBank/EMBL/DDBJ accession number for the 16S rRNA gene sequence of strain SJW1-29^T^ is MN596019. The GenBank/EMBL/DDBJ accession number for the whole-genome sequence of strain SJW1-29^T^ is WHLY00000000.

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
