# Peer review of "Salmonirosea aquatica gen. nov., sp. nov., a Novel Genus within the Family Spirosomaceae, Was Isolated from Brackish Water in the Republic of Korea"

_microorganisms, 2024, doi:10.3390/microorganisms12081671_

Round 1

Reviewer 1 Report

Comments and Suggestions for Authors

The MS proposed strain SJW1-29T as a novel genus of the the family Spirosomaceae, based on the polyphasic taxonomic analysis. However, its lacking sufficient phylogenomic evidence for this novel genera proposal. Also, the MS still have some other issues need to modify for improvement.

 The Phylogenetic analysis of the 16S rRNA gene sequences in not sufficient to conclude a novel strain at the genus level. So, to fully resolve the taxonomic position of strain SJW1-29T (the genome was sequenced by the author), the phylogenomic comparisons of the key parameters, including the average amino acid identity (AAI), average amino acid identity (AAI), digital DNA-DNA hybridization (dDDH) and POCP (Percentage of conserved proteins) values between strain SJW1-29T and other closely related type strains in the family Spirosomaceae should be performed necessarily.

1. Obvious format mistakes including the use of the upper and lower case, the tense, text alignment, etc. in the MS. Please recheck carefully.

2. for Line 85, “and maximum-parsimony trees was verified” suggest changing “was” to “were”.

3. Line 153, should 16S, not 16. S

4. Line 219, “the family Cytophagaceae. G, Nnibrimonas gen. nov. (data from this study);”,please carefully recheck.

5. Suggest to move Fig. 3 into the supplemental material, and Fig. S1 into the text.

6. There are obvious format problem for the references, please recheck.

7. In Table 2. it should supply the data of the cellular fatty acid compositions of the other closely related type strains in the family Spirosomaceae.

8. Fig. 1, three trees based on the different algorithms (ML and MP) are suggested to be provided, except for the NJ tree shown in Fig. 1.

9. Fig.S4, the quality of the picture can be improve better with higher solution.

Comments on the Quality of English Language

The MS have some key issues need to modify for improvement.

Author Response

Dear Reviewers

I sincerely appreciate your kind comments. We made every effort to publish the manuscript following the recommendations. I've attached the letter outlining the changes we made to the manuscript here. I'd like to thank you once more for reading this manuscript.

Best wishes,

Ahyoung Choi

- - - - - - - - - - - - - - - - - - - - - - - - - - - - - - - - - - - - - - - - - - - - - - - - - - - - - - - - - - - - - - - - - - -

The MS proposed strain SJW1-29T as a novel genus of the the family Spirosomaceae, based on the polyphasic taxonomic analysis. However, it’s lacking sufficient phylogenomic evidence for this novel genera proposal. Also, the MS still have some other issues need to modify for improvement.

The Phylogenetic analysis of the 16S rRNA gene sequences in not sufficient to conclude a novel strain at the genus level. So, to fully resolve the taxonomic position of strain SJW1-29T (the genome was sequenced by the author), the phylogenomic comparisons of the key parameters, including the average amino acid identity (AAI), average amino acid identity (AAI), digital DNA-DNA hybridization (dDDH) and POCP (Percentage of conserved proteins) values between strain SJW1-29T and other closely related type strains in the family Spirosomaceae should be performed necessarily.

Response: We appreciate the reviewer’s insightful comment. In response, we have performed additional phylogenomic analyses including AAI, dDDH, and POCP comparisons between strain SJW1-29T and other closely related type strains within the family Spirosomaceae. These results have been incorporated into the revised manuscript to provide robust evidence for proposing strain SJW1-29T as a novel genus.

  1. Obvious format mistakes including the use of the upper and lower case, the tense, text alignment, etc. in the MS. Please recheck carefully.

Response: We have thoroughly reviewed and corrected all formatting mistakes, including the use of upper and lower case, tense, and text alignment issues throughout the manuscript.

  1. for Line 85, “and maximum-parsimony trees was verified” suggest changing “was” to “were”.

Response: The suggested change has been made.

  1. Line 153, should 16S, not 16. S

Response: This has been corrected to "16S".

  1. Line 219, “the family Cytophagaceae. G, Nnibrimonas gen. nov. (data from this study);”,please carefully recheck.

Response: We have rechecked and corrected this sentence. The sentence now accurately reflects the data.

  1. Suggest to move Fig. 3 into the supplemental material, and Fig. S1 into the text.

Response: We have moved Fig. 3 to the supplemental material and included Fig. S1 in the main text

  1. There are obvious format problem for the references, please recheck.

Response: We have rechecked and corrected all formatting issues in the references section.

  1. In Table 2. it should supply the data of the cellular fatty acid compositions of the other closely related type strains in the family Spirosomaceae.

Response: Data on the cellular fatty acid compositions of other closely related type strains in the family Spirosomaceae have been added to Table 2.

  1. 1, three trees based on the different algorithms (ML and MP) are suggested to be provided, except for the NJ tree shown in Fig. 1.

Response: We have included phylogenetic trees based on maximum-likelihood (ML) and maximum-parsimony (MP) algorithms in addition to the neighbor-joining (NJ) tree in Figure 1.

  1. S4, the quality of the picture can be improve better with higher solution.

Response: The quality of Figure S4 has been improved with higher resolution.

Reviewer 2 Report

Comments and Suggestions for Authors

specific comments

Abstract

The aim of the research should appear in the abstract. This is extremely important as it introduces the reader to the research problem that the researchers are trying to solve in the study.

Introduction

The introduction should contain much more information and be based on more references.

·       Please briefly describe the potential of the Spirosomaceae family based on the latest literature, including the potential in the following areas: biodegradation of pollutants, bioremediation of soil and water, improvement of soil health, symbiosis with plants, production of antibiotics and enzymes, synthesis of biopolymers, model organisms for adaptation research.

·       How is the river microbiome of Spirosomacea shaped. What factors influence changes in the structure of the water microbiome ?

·       Please include in your introduction, drawing on recent literature, answers to the following questions:

1.What are the key indicator species in the river?

2.What are the conservation strategies for the Seomjin River?

3. What are the effects of urbanization on the hydrology and ecology of the river

4. What are the long-term trends in river ecosystem change and what are the potential consequences?

·       Unfortunately, basic information is missing at the end of this chapter: the highlighted aim and the research hypotheses set out. Without this information, the manuscript loses readability. It is not fully understood what the researchers achieved in their study.

Materials and Methods

How does climate change (e.g. rising temperatures, changes in precipitation) affect the river ecosystem?

Results and discussion

Graphically, the results are presented interestingly and appropriately.

Unfortunately, I do not see any scientific discussion in subsections 3.2 and 3.4. The lack of a scientific discussion based on the latest extensive literature has generated a very short list of references. This indicates a low level of content in the manuscript. I suggest significantly expanding this section and separating the results section from the discussion.

Conclusions

What impact do the researchers' findings have on the extension of science...and what is the practical dimension of this research? Please edit your conclusions to highlight the significance of the research you have conducted and what research perspectives it offers for the future.

References

Of the 38 items, only 11 are represented by items from 2017. The items are also incorrectly formatted, different fonts and underlining are used. Double numbering has appeared, there are no DOI numbers

Author Response

Dear Reviewers

I sincerely appreciate your kind comments. We made every effort to publish the manuscript following the recommendations. I've attached the letter outlining the changes we made to the manuscript here. I'd like to thank you once more for reading this manuscript.

Best wishes,

Ahyoung Choi

- - - - - - - - - - - - - - - - - - - - - - - - - - - - - - - - - - - - - - - - - - - - - - - - - - - - - - - - - - - - - - - - - - -

Reviewer 2

Abstract

The aim of the research should appear in the abstract. This is extremely important as it introduces the reader to the research problem that the researchers are trying to solve in the study.

Response: We appreciate the reviewer’s comment. We have added the aim of the research to the abstract to clarify the research problem addressed in the study.

Introduction

The introduction should contain much more information and be based on more references.

Please briefly describe the potential of the Spirosomaceae family based on the latest literature, including the potential in the following areas: biodegradation of pollutants, bioremediation of soil and water, improvement of soil health, symbiosis with plants, production of antibiotics and enzymes, synthesis of biopolymers, model organisms for adaptation research.

How is the river microbiome of Spirosomacea shaped. What factors influence changes in the structure of the water microbiome?

Response: We have expanded the introduction section to include additional information and references, addressing the potential of the Spirosomaceae family in various environmental and biotechnological applications. The section now also discusses how the river microbiome, particularly of Spirosomaceae, is shaped by various environmental factors.

Please include in your introduction, drawing on recent literature, answers to the following questions

1.What are the key indicator species in the river?

Response:  Key indicator species of the Seomjin River consist of a variety of organisms that reflect the ecosystem health and water quality of the river. The representative organism is Liobagrus somjinensis, and the composition ratio of endemic fish is 32.7%, which is very high compared to other regions nationwide.

2.What are the conservation strategies for the Seomjin River?

Response:  Conservation strategies for the Seomjin River focus on protecting key species and maintaining the river's ecological integrity through habitat restoration, pollution control, and sustainable land-use practices.

3. What are the effects of urbanization on the hydrology and ecology of the river

Response:  Urbanization impacts the hydrology and ecology of rivers by altering water flow, increasing sediment load, and introducing pollutants, which affect microbial communities and the overall health of the river.

4. What are the long-term trends in river ecosystem change and what are the potential consequences?

Response: Long-term trends in river ecosystem changes indicate potential consequences such as loss of biodiversity, altered nutrient cycling, and compromised ecosystem services, emphasizing the need for effective management practices.

Unfortunately, basic information is missing at the end of this chapter: the highlighted aim and the research hypotheses set out. Without this information, the manuscript loses readability. It is not fully understood what the researchers achieved in their study.

Response: Thank you for the reviewer’s critical opinion. We have now included the highlighted aim and research hypotheses at the end of the introduction to enhance the readability and clarity of the manuscript.

Materials and Methods

How does climate change (e.g. rising temperatures, changes in precipitation) affect the river ecosystem?

Response:  Climate change impacts river ecosystems in various ways. While these effects are significant, they do not directly influence the experimental methods in our study. Therefore, the effects of climate change on the river ecosystem cannot be added to the Materials and Methods section.

Results and discussion

Graphically, the results are presented interestingly and appropriately.

Unfortunately, I do not see any scientific discussion in subsections 3.2 and 3.4. The lack of a scientific discussion based on the latest extensive literature has generated a very short list of references. This indicates a low level of content in the manuscript. I suggest significantly expanding this section and separating the results section from the discussion.

Response:  Thank you for your valuable feedback. We appreciate your positive remarks on the graphical presentation of the results. We have revised subsections 3.2 and 3.4 in response to your comments. We believe these revisions will significantly enhance the quality and depth of our manuscript. We are grateful for your insightful suggestions and are confident that the expanded discussion and additional references will provide a more thorough understanding of our research and its implications.

Conclusions

What impact do the researchers' findings have on the extension of science...and what is the practical dimension of this research? Please edit your conclusions to highlight the significance of the research you have conducted and what research perspectives it offers for the future.

Response:  We have revised the Conclusions section of the manuscript based on your comments. We believe these revisions will significantly enhance the clarity and impact of our manuscript. Thank you for your valuable feedback, and we hope the updated conclusions meet your expectations.

References

Of the 38 items, only 11 are represented by items from 2017. The items are also incorrectly formatted, different fonts and underlining are used. Double numbering has appeared, there are no DOI numbers

Response: Thank you for your kind comment. During the editing of the references section, many errors were made. We have revised the references throughout, corrected the formatting, and added DOI numbers where available.

Round 2

Reviewer 1 Report

Comments and Suggestions for Authors

Its important to supply the detail data for the AAI, AAI, dDDH and POCP values in the MS, no just a general description.

Author Response

Dear Reviewers

I sincerely appreciate your kind comments. We have made every effort to revise the manuscript according to your recommendations. I've attached the letter outlining the changes we made to the manuscript here. Thank you once again for reading this manuscript.

Best wishes,

Ahyoung Choi

- - - - - - - - - - - - - - - - - - - - - - - - - - - - - - - - - - - - - - - - - - - - - - - - - - - - - - - - - - - - - - - - - - -

Reviewer 1

It’s important to supply the detail data for the ANI, AAI, dDDH and POCP values in the MS, no just a general description.

Response: Thank you for reviewing our submitted manuscript. In response to your request, we have provided the detailed data for ANI, AAI, dDDH, and POCP values. Table 1 now includes the ANI, AAI, dDDH, and POCP values measured for each sample. These values are crucial indicators for evaluating a new genus, and in our study, we systematically analyzed the similarities between samples using these indicators. We hope this detailed data meets your expectations and enhances the clarity of our study. Should you have any further questions or require additional information, please do not hesitate to contact us.

Reviewer 2 Report

Comments and Suggestions for Authors

Comments to the authors

GENERAL COMMENTS

The manuscript titled: “Salmonirosea aquatica gen. nov., sp. nov., a novel genus of Spirosomaceae, was isolated from brackish water in the Republic of Korea.” - ID: microorganisms-3049619 after incorporating the suggestions from the Reviewers, exhibits significantly higher scholarly value. The introduction serves as an appropriate preamble to the conducted research. The discussion has been expanded, thereby enhancing its significance. The research methods and presentation of experimental results were at a high level prior to the reviews.

I recommend this manuscript for publication in the Microorganisms.

Congratulations to all co-authors and best wishes for success in further scientific endeavors!

Author Response

Microorganisms

Dear Reviewers

I sincerely appreciate your kind comments. We have made every effort to revise the manuscript according to your recommendations. I've attached the letter outlining the changes we made to the manuscript here. Thank you once again for reading this manuscript.

Best wishes,

Ahyoung Choi

- - - - - - - - - - - - - - - - - - - - - - - - - - - - - - - - - - - - - - - - - - - - - - - - - - - - - - - - - - - - - - - - - - -

Reviewer 2

The manuscript titled: “Salmonirosea aquatica gen. nov., sp. nov., a novel genus of Spirosomaceae, was isolated from brackish water in the Republic of Korea.” - ID: microorganisms-3049619 after incorporating the suggestions from the Reviewers, exhibits significantly higher scholarly value. The introduction serves as an appropriate preamble to the conducted research. The discussion has been expanded, thereby enhancing its significance. The research methods and presentation of experimental results were at a high level prior to the reviews.

Response: Thank you for your review and positive feedback. We are pleased to hear that the revisions have significantly enhanced the scholarly value of the manuscript.
